# Effect of Gardeniae Fructus Powder on Growth Performance, Antioxidant Capacity, Intestinal Barrier Function, and Colonic Microbiota of Weaned Piglets

**DOI:** 10.3390/ani15020221

**Published:** 2025-01-15

**Authors:** Shilong Liu, Min Tian, Ming Ma, Yueqin Qiu, Jiaxi Tang, Jing Hou, Qi Lu, Chaoyang Tian, Guohao Ye, Li Wang, Kaiguo Gao, Shining Guo, Zongyong Jiang, Xuefen Yang

**Affiliations:** 1College of Veterinary Medicine, South China Agricultural University, Guangzhou 510642, China; liushilong94@126.com (S.L.); mm0002@163.com (M.M.); 2Institute of Animal Science, Guangdong Academy of Agricultural Sciences, Guangzhou 510640, China; tianmin@gdaas.cn (M.T.); qiuyueqin87@126.com (Y.Q.); tangjx0731@126.com (J.T.); houjing17370912304@163.com (J.H.); 2112209055@stu.fosu.edu.cn (Q.L.); 18851103382@163.com (C.T.); yghroy@163.com (G.Y.); wangli1@gdaas.cn (L.W.); gaokaiguo@gdaas.cn (K.G.); jiangzy@gdaas.cn (Z.J.); 3State Key Laboratory of Swine and Poultry Breeding Industry, Guangzhou 510640, China; 4Key Laboratory of Animal Nutrition and Feed Science in South China, Ministry of Agriculture and Rural Affairs, Guangzhou 510640, China; 5Guangdong Key Laboratory of Animal Breeding and Nutrition, Guangzhou 510640, China

**Keywords:** piglet, Gardeniae Fructus, antioxidant capacity, intestinal barrier function, colonic microbiota

## Abstract

Recent studies have demonstrated that Gardeniae Fructus (GF), its extracts, and active components (geniposide, genipin, and chlorogenic acid) alleviated colitis by improving intestinal antioxidant and immune capacity. However, little attention has been paid to the possible regulating effect and the underlying mechanism of GF powder on piglets suffering from weaning stress. This study investigated the effects of GF powder on the growth performance, diarrhea rate, antioxidant and immune capacity, and intestinal health of weaned piglets. The results show that GF supplementation at the 0.8% level significantly reduced the feed/weight gain (F/G) and diarrhea rate compared with a basal diet. GF supplementation also improved the abundance of *Prevotella* and *Prevotella copri* in the colon, increased antioxidant capacity, and alleviated inflammatory response, ultimately maintaining gut barrier function. This work provides further insights into the beneficial effects of GF in swine production.

## 1. Introduction

Weaning stress refers to the physiological and psychological challenges faced by piglets during the weaning process, including diet changes, separation from the sow, and environmental adjustments [1]. In modern intensive animal husbandry, early weaning practices have further intensified these stressors. Extensive research has shown that weaning stress can lead to adverse effects in piglets, including growth delays, reduced feed intake, and an increased incidence of diarrhea [2,3,4,5]. The gastrointestinal tract, which serves as a critical digestive and immune organ, constitutes the first line of defense against external pathogens. Its health is paramount for the overall growth and well-being of piglets, particularly during the weaning phase [6]. Previous research has indicated that weaning stress negatively impacts the gut of piglets, manifesting as decreased villus height, increased crypt depth, heightened gut permeability, reduced digestive enzyme activity, and an imbalance in gut microbiota [7,8,9]. Therefore, effective improvement of the gut health of weaned piglets is essential to promote their healthy growth and improve breeding efficiency.

*Gardenia jasminoides* Ellis, commonly known as *Gardenia*, is one of approximately 250 species of flowering plants in the Rubiaceae family [10]. The dried mature fruit of this plant, known as Gardeniae Fructus (GF), is referred to as Fructus Gardeniae or Zhi-zi (ZZ) in China [11]. GF has a long history of use as a natural yellow dye in food and as a medicinal herb or tea in countries of Northeast Asia, including Korea, China, and Japan [12,13]. In particular, GF is one of the homologies of medicine and food [14]. Modern phytochemical analyses have revealed that GF contains four main types of chemical components in relatively high concentrations: iridoids, terpenoids, organic acids, and flavonoids [15]. The predominant iridoids in GF include geniposide, gardenoside, and geniposidic acid, and geniposide serves as a critical marker of the quality of GF [16]. Diterpenes in GF are mainly composed of crocin compounds, most of which are pigments. Among the organic acids contained in gardenia, chlorogenic acid has the highest content [16]. Furthermore, a previous study revealed that GF is rich in various types of pectin, including water-soluble pectin, acid-soluble pectin, xylan/alkaline-soluble pectin, and cellulose [17].

Recent studies have shown that GF, its extracts, and active components exhibit multiple physiological functions, including anti-inflammatory, antioxidant, antibacterial, and antiviral properties. In a study involving mice with TNBS-induced colitis, administration of GF ethanol extract for 7 days resulted in reduced Disease Activity Index (DAI) scores, improved intestinal morphology, and decreased expression levels of myeloperoxidase (MPO), nitric oxide (NO), malondialdehyde (MDA), Interleukin-1β (IL-1β), tumor necrosis factor-α (TNF-α), and Interleukin-6 (IL-6) in the colon [18]. In the TNBS-induced mouse colitis model, geniposide was found to alleviate colitis by reducing the release of pro-inflammatory cytokines and restoring the integrity of the damaged intestinal barrier [19]. Furthermore, in DSS-induced colitis mice, geniposide enhances the antioxidant capacity of the colon by activating the Nrf2/HO-1 signaling pathway while inhibiting the expression of pro-inflammatory factors [20]. Another active component, crocin, has shown potential in treating acute myocardial infarction. In a mouse model of myocardial infarction, pretreatment with crocin reduced serum creatine kinase (CK) and malondialdehyde (MDA), as well as lactate dehydrogenase (LDH) and superoxide dismutase (SOD). The mechanism of action is believed to involve inhibition of the Rho/ROCK/NF-κB pathway [21]. Despite these promising findings, research on the application of GF in piglets remains relatively limited.

Thus, this study aims to explore the effects of Gardeniae Fructus powder on growth performance, diarrhea rate, antioxidant capacity, immune function, and intestinal health of weaned piglets. By investigating these parameters, we seek to evaluate the potential of GF as a natural feed additive that could mitigate the negative impacts of weaning stress and promote overall piglet health. This research is particularly relevant given the increasing interest in natural alternatives to conventional growth promoters and the limited existing research on the application of GF in swine nutrition.

## 2. Materials and Methods

Animal protocols in the present study were performed following the Guidelines for the Care and Use of Animals for Research and Teaching following approval by the Animal Care and Use Committee of Guangdong Academy of Agricultural Science (authorization number GAASIAS-2016-017).

### 2.1. Animals, Diets and Management

A total of 144 [Duroc × (Landrace × Yorkshire)] weaned piglets (8.29 ± 0.11 kg) at 21 d old were randomly assigned to 4 groups, including a basal diet supplement with 0%, 0.4%, 0.6%, and 0.8% Gardeniae Fructus powder, respectively (n = 36). Each treatment consisted of 6 replicate pens, with 6 piglets per pen and each pen containing 3 barrows and 3 gilts. The piglets were fed. The experimental period was 30 days. The basal diet (control) was formulated to meet the nutrient recommendations of the National Research Council 2012 (NRC, 2012) (Table 1) [22]. All the pigs were allotted slatted floors (1.8 × 2.5 m) and raised at the experimental farm of the Institute of Animal Science, Guangdong Academy of Agricultural Sciences, China. The temperature-controlled nursery house temperature for the first week was set at 28 ± 1 °C, with a daily decrease of 0.5 °C starting from the second week until 25 °C was reached. Each pen was equipped with a one-sided feeder and two stainless steel nipple drinkers. The piglets were fed four times a day (at 8:00 a.m., 12:00 p.m., 4:00 p.m. and 8:00 p.m.) with the prepared diet in feeding troughs and had ad libitum access to feed and water. At the beginning and end of the experiment, all experimental pigs were weighed on an empty stomach on an individual basis. During the experiment, the feed intake of each replicate was accurately recorded, and the diarrhea situation of the pigs was recorded to calculate the average daily weight gain, average daily feed intake, feed-to-weight ratio, and diarrhea rate. Two observations per day (at 9:00 a.m. and 5:00 p.m.) recorded defecation status/stool condition. The stool-scoring system was as follows: 0 points—formed, cylindrical, or pellet-like stool (normal consistency); 1 point—soft stool that maintains its shape; 2 points—thick/pasty stool, shapeless, with no separation between solid and liquid; 3 points—liquid stool, shapeless, with separation between solid and liquid. A stool score ≥ 2 points is defined as diarrhea.

The dried mature fruit of *Gardenia jasminoides* is rarely elliptical, measuring 2.5–3.5 cm in length and 1–1.5 cm in diameter, and is deep red or reddish yellow. The dried mature fruit was powdered using a pulverizer and filtered through a sieve (80 mesh), then stored at room temperature (25 °C) for use. As in our previous study, the content of crude protein, crude fat, crude fiber, crude ash, alcium, and total phosphorus were determined based on the conventional nutrient determination methods of feed ingredients, adopting the national standard method of GB/T6432-2014 (https://www.chinesestandard.net/Default.aspx?StdID=GB/T6432%e2%80%942014, accessed on 9 December 2024), GB/T6433-2014 (https://www.chinesestandard.net/PDF/English.aspx/GBT6433-2006?Redirect, accessed on 9 December 2024), GB/T6434-2014 (https://www.chinesestandard.net/PDF/English.aspx/GBT6434-2022?Redirect, accessed on 9 December 2024), GB/T6438-2014 (https://www.chinesestandard.net/PDF/English.aspx/GBT6438-2007?Redirect, accessed on 9 December 2024), GB/T6436-2018 (https://www.chinesestandard.net/PDF/English.aspx/GBT6436-2018?Redirect, accessed on 9 December 2024), and GB/T6437-2018 (https://www.chinesestandard.net/PDF/English.aspx/GBT6437-2018?Redirect, accessed on 9 December 2024) [23]. The main active ingredients of the GF powder, namely total flavonoids, total phenol, and soluble sugar, were determined based on the plant flavonoids test kit (A142-1-1), plant total phenol test kit (A143-1-1) and plant soluble sugar content test kit (A145-1-1) (Nanjing Jiancheng Institute of Bioengineering, Nanjing, China). The nutritional levels and main bioactive compounds of GF powder are listed in Appendix A.

### 2.2. Sample Collections

At 8 a.m. on the 31st day, one piglet was chosen from each pen based on average body weight. First, blood was collected from the anterior vena cava of these piglets, and then these piglets were euthanized with pentobarbital sodium (10 mg/kg) and slaughtered. Serum samples were left at room temperature for 30 min, then centrifuged at 3000 r/min for 15 min and stored at −20 °C. Segments and mucosal samples from the middle jejunum and distal ileum were rinsed with ice-cold PBS and then fixed in 4% paraformaldehyde or liquid nitrogen. The digesta from the colon was immediately collected and frozen in liquid nitrogen and stored at −80 °C with intestinal mucosal samples.

### 2.3. Serum Biochemical Parameters

Serum biochemical indicators were detected using the VITAL fully automated biochemical analyzer (SELECTA ProXL), all using the Zhongsheng Beikong Biotechnology Co., Ltd. reagent kit (Hefei, China), and determined according to the kit instructions. Determination of serum samples was undertaken as follows: blood creatinine (CRE, picric acid method, item number: 100020170), urea (UREA, urease glutamate dehydrogenase method, item number: 100000280), blood glucose (GLU, glucose oxidase peroxidase method, item number: 200891), triglycerides (TG, glycerophosphate oxidase peroxidase method, item number: 198021), total protein (TP, biuret method, item number: 180531), albumin (Alb, bromocresol green method, item number: 180641), high-density lipoprotein (HDL-C, direct peroxidase method, item number: 100020235), and low-density lipoprotein (LDL-C, direct surfactant scavenging method, item number: 100020245).

### 2.4. Serum Immunological Indicators

Serum inflammation-related cytokines Interleukin 1β (IL-1β, Catalog No. MM-042201), Interleukin 6 (IL-6, Catalog No. MM025981), Interleukin 10 (IL-10, Catalog No. MM-042501), Interleukin 22 (IL-22, Catalog No. MM-123101), Tumor Necrosis Factor α (TNF-α, Catalog No. MM-038301), Transforming Growth Factor-β (TGF-β, Catalog No. MM-7791901), Serum Immunoglobulin A (IgA, Catalog No. MM-090501), Immunoglobulin G (IgG, Catalog No. MM-040301), Immunoglobulin M (IgM, Catalog No. MM-040201), and diamine oxidase (DAO, Catalog No.: ML002413) were all tested using Jiangsu Meimian Industrial Co., Ltd. kits (Nanjing, China), according to the instructions provided with the kits.

### 2.5. Antioxidant-Related Indicators

The antioxidant parameters were measured using commercially available kits (Nanjing Jiancheng Institute of Bioengineering, Nanjing, China) according to the manufacturer’s instructions. The total antioxidant capacity (T-AOC) was determined using the ABTS method (Catalog No.: A015-2-1). Malondialdehyde (MDA) levels were evaluated using the TBA method (Catalog No.: A003-2). Catalase (CAT) activity was measured using the ammonium molybdate method (Catalog No.: A007-1-1). Total superoxide dismutase (T-SOD) activity was evaluated using the hydroxylamine method (Catalog No.: A001-1). Glutathione peroxidase (GSH-Px) activity was determined using the dithiodinitrobenzoic acid method (Catalog No.: A005). Jejunal and ileal mucosa samples were homogenized in physiological saline to obtain 10% and 40% homogenates, respectively. The homogenates were then centrifuged, and the supernatants were used for subsequent analyses. The protein content in the jejunal and ileal mucosa homogenates was determined using a BCA protein assay kit (Catalog No.: 23227, Thermo Scientific, Waltham, MA, USA). Tissue antioxidant activities were normalized to protein content.

### 2.6. Intestinal Mucosal Morphology

Segments of approximately 2.0 cm in length were excised from the middle jejunum and distal ileum. These samples were rinsed with ice-cold PBS and fixed in 4% paraformaldehyde for morphometric evaluation and histochemical staining. Fixed samples were embedded in paraffin, and 4-μm cross-sections were mounted on polylysine-coated slides. The sections were then deparaffinized, rehydrated, and stained with hematoxylin and eosin (H&E) for intestinal morphological examination. H&E-stained sections were scanned using a digital brightfield microscope scanner (Pannoramic 250, 3D HISTECH, Budapest, Hungary). For each intestinal segment, 20 well-orientated and intact villi with adjacent crypts were randomly selected for measurement. The height and crypt depth were determined using slide viewer software (Case Viewer 2.3, 3D HISTECH, Budapest, Hungary). The height-to-crypt depth ratio of the villus was subsequently calculated.

### 2.7. qRT-PCR

Total RNA was extracted from jejunal and ileal mucosa samples using the Total RNA Extraction Kit (Invitrogen, Carlsbad, CA, USA) and dissolved in DEPC water, then stored at −80 °C. The concentration and purity of RNA were evaluated using a NanoDrop ND-1000 spectrophotometer (Bio-Rad, Hercules, CA, USA), and all samples showed an A260/A280 ratio of 1.8 to 2.0. RNA integrity was verified using 1% agarose gel electrophoresis. An equal amount (1 μg) of total RNA was used to synthesize the first-strand cDNA with a reverse transcription kit (Takara, Tokyo, Japan). The reverse transcription products were stored at −20 °C for later use. The qRT-PCR reaction (Bio-Rad CFX system) was set up in a 20 μL volume: 10.0 μL of iTaq Universal SYBR Green Supermix (Bio-Rad, Hercules, CA, USA), 1.0 μL each of the forward and reverse primers (10 μM/L), 2.0 μL of the cDNA template (10 times diluted) and 6.0 μL of ddH2O, with each sample run in triplicate. The amplification protocol was as follows: initial denaturation at 95.0 °C for 30 s; denaturation at 95.0 °C for 15 s; annealing for 30 s; extension at 72.0 °C for 30 s for 40 cycles, followed by melt-curve analysis. Primers were designed based on GenBank-derived pig gene sequences, targeting the conserved regions, and were synthesized by Shanghai Sangon Biotech Co., Ltd. (Shanghai, China). The primer sequences are listed in Table 2. The β-actin gene was used as an internal reference, and the relative expression levels of target genes mRNA were calculated using the 2^−ΔΔCt^ method.

### 2.8. Colonic Microbiome

Genomic DNA was extracted from colonic content samples using a QIAamp DNA kit (Qiagen, Hilden, Germany). The quality of DNA was assessed by electrophoresis on 1% agarose gels. The V3-V4 region of the 16S rRNA gene was amplified by PCR using sample-specific barcode primers for all colonic digesta samples. PCR reactions were performed in a total volume of 30 µL. The amplification products were purified using an Ion Plus Fragment Library Kit 48 rxns (Thermo Scientific, Waltham, MA, USA) to construct the library, and the library quality was evaluated using a Qubit 2.0 fluorometer (Thermo Scientific). Low-quality reads were trimmed using Cutadapt (v1.9.1). Raw reads were processed by removing barcodes and primer sequences. Chimeric sequences were eliminated by comparing processed reads against a species annotation database, resulting in clean reads. Operational taxonomic units (OTU) were delineated from the clean reads using UPARSE software (v7.0.1001). The diversity of the microbial community was analyzed using QIIME software (v1.9.1). Alpha diversity metrics, including observed_species, Shannon, Simpson, Chao1, ACE, and PD_whole_tree indices, were used to assess diversity within samples. Beta diversity, representing diversity between samples, was evaluated using principal component analysis (PCA), principal coordinate analysis (PCoA), and nonmetric multidimensional scaling (NMDS).

### 2.9. Statistical Analysis

Data were analyzed using SPSS 20.0 (SPSS Inc., Chicago, IL, USA) and presented as mean ± standard error (SEM). The results were assessed using a one-way analysis of variance (ANOVA) followed by Duncan’s test. Statistical significance was established at *p* < 0.05.

## 3. Results

### 3.1. Effect of GF on Growth Performance of Weaned Piglets

As shown in Table 3, our results indicated that a dietary supplement with *Gardeniae Fructus* powder at 0.8% level significantly decreased F/G and diarrhea rate compared to piglets fed with a basic diet at 0% level (*p* < 0.05). Furthermore, final weight, ADG and ADFI had no significant difference in the four groups (*p* < 0.05).

### 3.2. Effect of GF on Serum Biochemical Parameters of Weaned Piglets

As shown in Table 4, compared with basal diet, supplements with 0.6 and 0.8% GF significantly increased HDL-C content in serum (*p* < 0.05). In addition, the contents of CRE, UREA, GLU, TG, TP, ALB, LDL-C and DAO had no significant changes in the four groups (*p* > 0.05).

### 3.3. Effect of GF on Serum Immunological Indicators of Weaned Piglets

In Table 5, our results showed that the serum’s IL-10 content remarkably increased in GF supplement groups compared with basal diet (*p* < 0.05). At the same time, in comparison with the basal diet, 0.8% GF significantly increased the contents of IL-6 and Ig A (*p* < 0.05); 0.4% GF significantly increased the contents of Ig G in serum (*p* < 0.05). Compared with 0.4% GF, 0.6% GF significantly increased the contents of IL-1β in serum (*p* < 0.05). The results also indicated that serums IL-22, TNF-α, TGF-β, and Ig M contents had no significant difference in the current study (*p* > 0.05).

### 3.4. Effect of GF on Antioxidant-Related Indicators in Serum and Intestine of Weaned Piglets

As shown in Table 6, compared with basal diet, serum MDA content significantly decreased in 0.4% GF and 0.8% GF groups as well as T-SOD content significantly increased (*p* < 0.05). T-AOC content significantly increased in 0%, 0.4% and 0.8% GF groups compared with basal diet (*p* < 0.05). Additionally, there is no significant difference in serum CAT and GSH-Px contents in four groups (*p* > 0.05).

Then, we determined the antioxidant enzyme activity and related gene expression in the intestinal mucosa. In Table 6 and Figure 1, supplement with GF at 0.4%, 0.6% and 0.8% level significantly decreased MDA content in jejunal and ileal mucosa compared with basal diet (*p* < 0.05). Our results showed that GSH-Px content significantly increased in 0.8% GF compared with the other three groups in jejunal mucosa (*p* < 0.05); T-AOC content significantly increased in 0.4% GF compared with the other three groups in ileal mucosa (*p* < 0.05). Compared with basal diet, T-AOC and CAT content in ileal mucosa were significantly increased in 0.4% and 0.8% GF (*p* < 0.05).

In the jejunal mucosa, compared with basal diet, 0.4% GF significantly increased *Nrf 2*, *NQO*, *SOD 1*, and *GCLC* gene expression (*p* < 0.05), 0.8% GF significantly increased *SOD 1* and *GCLM* gene expression (*p* < 0.05). In the ileal mucosa, 0.4% GF significantly increased *Nrf 2*, *HO-1* and *GCLC* gene expression (*p* < 0.05), 0.6% GF significantly increased *Nrf 2* gene expression (*p* < 0.05), 0.8% GF significantly increased *Nrf 2*, *NQO*, and *GCLM* gene expression (*p* < 0.05).

### 3.5. Effect of GF on Intestinal Mucosal Morphology of Weaned Piglets

As shown in Table 7 and Figure 2, the present results indicated that supplements with GF at 0.8% level significantly increased villus height in the jejunum (*p* < 0.05) and tended to increase villus/crypt ratio compared with basal diet (*p* = 0.057). In the ileum, compared with the basal diet, 0.6% and 0.8% GF significantly increased villus height, along with a 0.8% GF increase in the villus/crypt ratio (*p* < 0.05).

### 3.6. Effect of GF on Expression of Genes Related to Intestinal Epithelium Functions in Weaned Piglets

As shown in Figure 3, compared with basal diet, 0.4% GF significantly increased *Occludin* gene expression in ileal mucosa (*p* < 0.05), 0.6% GF significantly increased *ZO-1*, *Claudin-1* and *Occludin* gene expression in jejunal mucosa (*p* < 0.05), 0.8% GF significantly increased *ZO-1*, *Occludin* gene expression in jejunal mucosa along with *Occludin* expression in ileal mucosa (*p* < 0.05).

### 3.7. Effect of GF on Colonic Microbiome Composition of Weaned Piglets

As shown in Figure 4a, there were 661 OTUs among the four groups. The results of α-diversity indicated that 0.8% GF significantly increased Shannon (Figure 4b), Observed species (Figure 4e) and Chao 1 index (Figure 4f) compared with basal diet (*p* < 0.05). 0.6% and 0.8% GF significantly decreased the Goods coverage index compared with basal diet (Figure 4d) (*p* < 0.05). PCoA (Figure 4g) and NMDS (Figure 4h) showed the β-diversity of colonic microbiota.

The relative abundance of colonic microbiota is shown in Figure 5. At the phylum level (Figure 5a), the 10 most prevalent microbes were *Firmicutes*, *Bacteroidetes*, *Actinobacteria*, *Proteobacteria*, *Spirochaetes*, *Tenericutes*, *WPS-2*, *Cyanobacteria, Fibrobacteres*, and *Verrucomicrobia*. At the genus level (Figure 5d), the 10 most prevalent microbes were *Lactobacillus*, *Prevotella*, *Blautia*, *Roseburia*, *Collinsella*, *Gemmiger*, *Megasphaera*, *Coprococcus*, *Oscillospira*, and *Faecalibacterium*. At the species level (Figure 5g), the 20 most prevalent microbes were *Prevotella copri*, *Lactobacillus reuteri*, *Lactobacillus salivarius*, *Gemmiger formicilis*, *Collinsella aerofaciens*, *Roseburia faecis*, *Lactobacillus mucosae*, *Lactobacillus helveticus*, *Faecalibacterium prausnitzii*, *Prevotella stercorea*, *Eubacterium biforme*, *Butyricicoccus pullicaecorum*, *Lactobacillus delbrueckii*, *Coprococcus catus*, *Dorea formicigenerans*, *Blautia obeum*, *Bulleidia p-1630-c5*, *Ruminococcus bromii*, *Ruminococcus flavefaciens*, and *Lactobacillus pontis*.

Our results indicated that in comparison with the basal diet, the relative abundance of *Firmicutes* significantly decreased in 0.4%, 0.6%, and 0.8% GF groups as well as *Bacteroidetes* increased in 0.8% GF group at the phylum level (*p* < 0.05) (Figure 5b,c). At the genus level, compared with the basal diet, 0.6% and 0.8% GF significantly increased *Prevotella* abundance, and 0.6% GF significantly decreased *Coprococcus* abundance (*p* < 0.05) (Figure 5e,f). At the species level, compared with basal diet, 0.8% GF significantly increased *Prevotella copri* abundance, and 0.4%, 0.6%, and 0.8% GF significantly decreased *Blautia obeum* abundance (*p* < 0.05) (Figure 5h,i).

### 3.8. The Spearman Correlation Analysis Between Colonic Microbiota and the Performance

As shown in Figure 6, the canonical correlation analysis between significantly differential colonic microbiota and antioxidant-related indicators, intestinal epithelium functions. Our results showed that ileal villus height, serum T-AOC content, ileal CAT, and T-AOC content expressions of *NQO*, *GCLC* and *GCLM* were negatively associated with the abundance of *Firmicutes* (phylum) (*p* < 0.05). Jejunal and ileal villus heights were negatively associated with the abundance of *Coprococcus* (genus) (*p* < 0.05). Ileal villus height was negatively associated with the abundance of *Blautia obeum* (species) (*p* < 0.05). Ileal villus height, ileal CAT content, expressions of jejunal *occluding*, ileal *NQO, GCLC* and *GCLM* were positively associated with the abundance of *Bacteroidetes* (phylum) and *Prevotella* (genus) (*p* < 0.05). Ileal villus height, expressions of jejunal *Claudin-1* and *Occludin*, ileal *GCLC* and *GCLM* were positively associated with the abundance of *Prevotella copri* (species) (*p* < 0.05).

## 4. Discussion

Weaning stress in piglets typically manifests itself as decreased growth performance and an increased incidence of diarrhea immediately after abrupt weaning [24,25]. The gene and protein expression of tight junction proteins (Claudin-1, Occludin, and ZO-1) are significantly reduced after weaning of piglets [26]. Zhu et al. found that weaning stress led to an increase in serum levels of nitric oxide and hydrogen peroxide, a decrease in the expression of antioxidant enzymes in the jejunum, and damage to intestinal villi in piglets [27]. Previous studies have shown that plant extracts (such as Guizhi Li-Zhong extract, *Huangqin* decoction, *Codonopsis Pilosulae* and *Astragalus Membranaceus* extract) can alleviate oxidative stress, improve immunity capacity, increase beneficial bacteria abundance, and improve intestinal barrier function, therefore alleviating weaning stress in piglets [28,29,30]. A study by Liu et al. indicated that *Gardenia jasminoides* Ellis fruit extracts relieved oxidative stress and inflammatory reaction, therefore alleviating TNBS-induced colitis in rats [18]. In the present study, our results indicated that dietary supplements with *Gardeniae Fructus* powder at a level of 0.8% significantly decreased piglet diarrhea rate and F/G after a 30-day feeding trial. At the same time, there is an increasing trend for ADG in the 0.8% GF group. Supplements with *Gardeniae Fructus* powder (0.8%) significantly improved the levels of T-AOC, T-SOD, CAT, and IL-10 and decreased MDA in serum and small intestine compared with the control group, therefore increasing villus height in the ileum. The indices related to intestinal barrier function showed GF significantly increased expression of *ZO-1*, *Occludin* and *Claudin-1*, which may be associated with the improvement of *Prevotella* and *Prevotella copri* relative abundance in piglets. Thus, in the present study, GF reduced the diarrhea rate and F/G, maybe by increasing intestinal antioxidant levels, villus height, and intestinal barrier function in piglets.

Serum biochemical indicators serve as crucial tools for assessing an organism’s health status, providing valuable insights into the functioning of vital organs and metabolic processes. These indicators can reflect the function and metabolic state of multiple organ systems [31]. Cytokine factors are crucial for preserving bodily health, responding to diseases, and modulating various physiological processes. Their impact extends beyond conventional inflammatory reactions, encompassing a broad spectrum of physiological and pathological mechanisms [32,33]. In the present study, our results showed that the serum’s IL-10 content remarkably increased in GF supplement groups compared with the basal diet. At the same time, in comparison with the basal diet, the contents of Ig G, IL-6 and Ig A were significantly increased in serum at 0.4% and 0.8% GF levels, respectively. As stated in previous studies, the cytokines of the IL-10 family exert essential functions to maintain tissue homeostasis during infection and inflammation by restricting excessive inflammatory responses [34]. IgG, IgM, and IgA are the primary immunoglobulins that protect animals from infection [35]. According to the results, we speculated that dietary supplementation with GF improved piglet immunity by increasing serum immunoglobulin and anti-inflammatory factor levels.

Weaning stress is closely associated with the onset of oxidative stress [36]. Previous studies have demonstrated that oxidative stress affects intestinal barrier function [37] and compromises immunity [38]. A complex network of enzymes protects organisms from oxidative damage by maintaining a balance between oxidative and antioxidant defense systems. This balance is reflected in blood biomarkers, with increased oxidative indicators (such as MDA) and decreased antioxidant indicators (such as GPx and SOD) [39]. When reactive oxygen species (ROS) accumulate due to insufficient degradation, they induce lipid peroxidation of fatty acids, forming malondialdehyde (MDA). In our study, we found that supplementation with GF significantly reduced MDA content and increased T-SOD, T-AOC, CAT, and GSH-Px contents in serum, jejunum, and ileum. The results of gene expression revealed that *Nrf-2*, *HO-1*, *NQO*, *SOD1*, *GCLM*, and *GCLC* increased in jejunum and ileum after supplementing GF in the diet. These findings are consistent with previous research showing that dietary supplementation with alfalfa meal decreased serum MDA and ROS concentrations, alleviating weaning stress and improving early intestinal function [40]. Furthermore, geniposide, a compound derived from gardenia fruit, has been shown to ameliorate inflammation and oxidative stress in experimental colitis by modulating the Nrf-2/HO-1/NF-κB pathway [20]. Therefore, our results suggest that GF reduces lipid peroxidation products and protects cells from oxidative stress.

The height of villi and the depth of crypts are both indicators of intestinal mucosal development and reflect the digestive and absorptive capacity of the intestine. The present results showed that supplement with GF at 0.8% level significantly increased villus height in jejunum, villus height and villus/crypt ratio in ileum compared with basal diet. This suggests that GF improved the digestive ability of the small intestine. The intestinal epithelial barrier comprises intestinal epithelial cells and tight intercellular junctions, including proteins such as ZO-1, Occludin, and Claudin-1 [26]. Weaning stress can induce intestinal epithelial cells and suppress the expression of tight junction proteins, ultimately increasing the permeability of the intestinal epithelial barrier [41]. In our study, we observed a significantly increased expression of *ZO-1*, *Occludin*, and *Claudin-1* in the GF groups. This finding is consistent with previous research showing that geniposide (purified from *Gardenia jasminoides* Ellis) increased the expression of tight junction proteins and improved TNBS-induced rat colitis [19]. These results support our experimental findings that GF enhances intestinal barrier function by promoting the expression of tight junction proteins.

The intestinal microbiota can be considered an additional organ for the host, as a stable and healthy microbial community supports the host’s growth and overall health [42,43,44]. In piglets, the colonic microbiota plays a crucial role in the breakdown of complex carbohydrates and the production of short-chain fatty acids, such as butyrate, which serve as an energy source for intestinal epithelial cells. Moreover, these microorganisms contribute to vitamin synthesis and enhance mineral absorption [45]. Our study showed that GF supplementation significantly altered the microbial composition. At the phylum level, we observed a decrease in *Firmicutes* and an increase in *Bacteroidetes*. At the genus level, the relative abundance of *Prevotella* increased in 0.4% and 0.8% GF groups. *Prevotella* is one of the most predominant genera in the large intestine of pigs [46,47]. The *Prevotella*-driven enterotype has been positively associated with several important animal traits, including feed intake [48], feed efficiency [49], weight gain [50] and the incidence of diarrhea [51]. These findings suggest that *Prevotella* plays an important role in mediating growth performance and disease resilience in pigs. Interestingly, in ruminants, *Prevotella* in the rumen has been shown to promote carbohydrate utilization and enhance nitrogen metabolism [52,53]. However, in humans, as one of the most abundant bacteria in the intestine, an increase in *Prevotella* abundance is positively correlated with the occurrence of inflammation, arthritis, and other conditions [54,55].

At the species level, the abundance of *Blautia obeum* decreased while *Prevotella copri* increased. The *Prevotella* genus encompasses more than 40 different culturable species, of which three, *Prevotella copri*, *Prevotella salivae*, and *Prevotella stercorea*, are members of the gut microbiota [56]. *Prevotella copri* is the best studied and abundant intestinal species in the genus *Prevotella* genus [57]. One of the key reasons for its abundance in the human gut is the preferential metabolism of xylan, a plant polysaccharide found in plant-based diets, by this species [58]. Similar to *Prevotella*, *Prevotella copri* has also shown controversial results in different models. For example, a recent study linked higher intestinal abundance of *Prevotella copri* to rheumatoid arthritis [59]. Another study showed that maternal carriage of *Prevotella copri* during pregnancy reduces the offspring’s risk of allergic disease in the offspring through the production of succinate [60]. A study by Shen et al. indicated that the combination of *Prevotella copri* and *L. murinus* inhibits the TGF-β1/Smad pathway and reduces inflammation and fibrosis in primary sclerosing cholangitis [61]. The strange discrepancies among *Prevotella copri* studies have only recently been attributed to the diversity of its strains, which differ substantially in their encoded metabolic patterns from the commonly used reference strain [57]. Despite inconsistent conclusions about the role of bacteria, one point is widely accepted: bacteria can effectively utilize plant-derived polysaccharides. *Prevotella copri* has unique PULs, which vary between different strains, and its gene products can distinctively break down a wide range of plant-derived (but not animal-derived) polysaccharides in the human gut [62]. When metabolizing fiber, *Prevotella copri* produces short-chain fatty acids (SCFAs), which may be the reason *Prevotella copri* can protect the mucosal barrier and reduce inflammation [63].

## 5. Conclusions

In conclusion, dietary supplementation with 0.8% Gardeniae Fructus powder significantly decreased F/G and diarrhea rate and improved antioxidant capacity and intestinal barrier function, which may be associated with the improvement of *Prevotella copri* relative abundance. These findings indicate that Gardeniae Fructus powder may be used as a feed additive in swine weaning.

## Figures and Tables

**Figure 1 animals-15-00221-f001:**
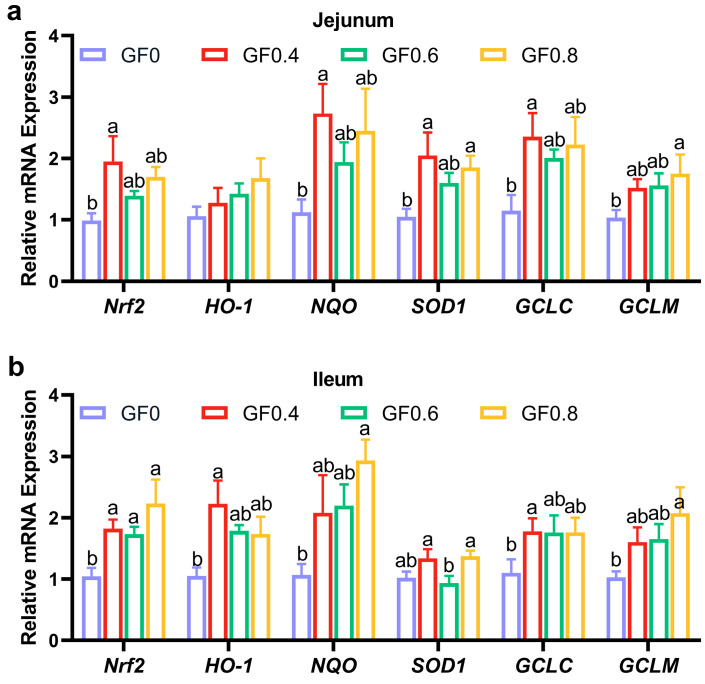
Effect of GF on antioxidant-related mRNA expression in the intestine of piglets. (**a**,**b**) mRNA expression in jejunum and ileum mucosa. ^a, b^ Means within a row with no common superscript differ significantly (*p* < 0.05) (n = 6). Nrf 2, Nuclear Factor Erythroid 2-Related Factor 2; HO-1, Heme Oxygenase-1; NQO, NAD(P)H Quinone Dehydrogenase 1; SOD 1, Superoxide Dismutase 1; GCLC, Glutamate-Cysteine Ligase Catalytic Subunit; GCLM, Glutamate-Cysteine Ligase Modifier Subunit. GF0, basal diet group; GF0.4, 0.4% GF group; GF0.6, 0.6% GF group; GF0.8, 0.8% GF group.

**Figure 2 animals-15-00221-f002:**
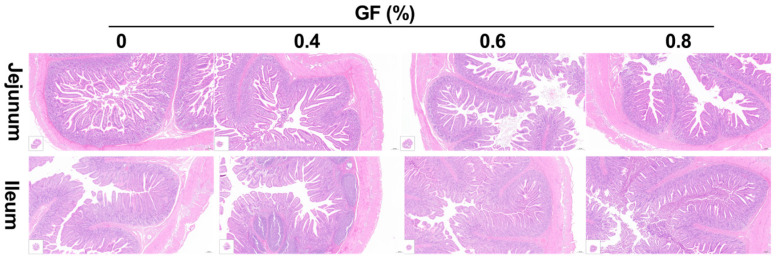
Effect of GF on intestinal mucosal morphology of weaned piglets (n = 6). The scale of the figure is 200×.

**Figure 3 animals-15-00221-f003:**
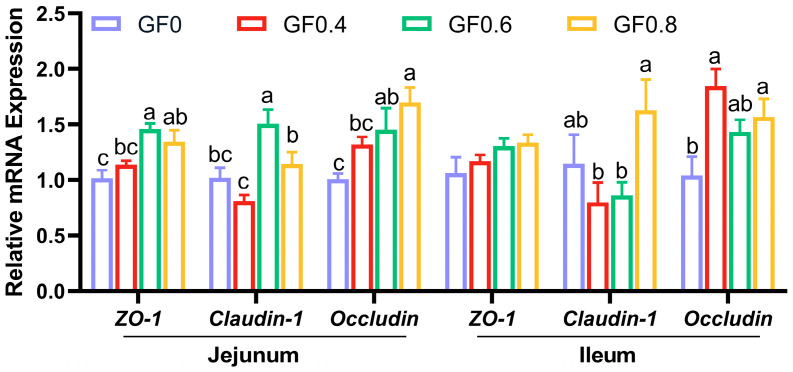
Effect of GF on mRNA expression of intestinal epithelium functions in the jejunum of piglets. ^a,b,c^ Means within a row with no common superscript differ significantly (*p* < 0.05) (n = 6). ZO-1, zonula occludens-1. GF0, basal diet group; GF0.4, 0.4% GF group; GF0.6, 0.6% GF group; GF0.8, 0.8% GF group.

**Figure 4 animals-15-00221-f004:**
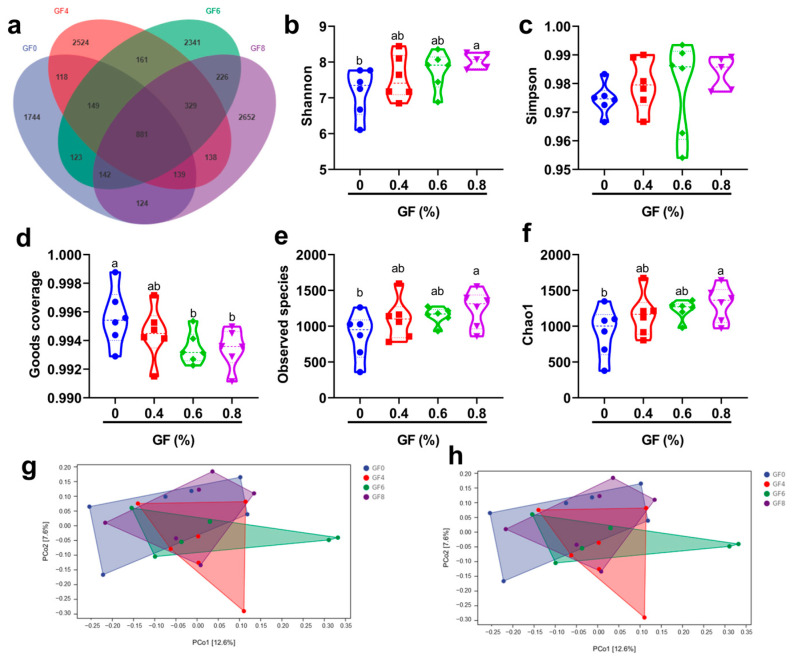
Effect of GF on α- and β-diversity of colonic microbiome in weaned piglets. ^a,b^ Means within a row with no common superscript differ significantly (*p* < 0.05) (n = 6). Venn diagram of OTU (**a**), Shannon index (**b**), Simpson index (**c**), Goods coverage index (**d**), Observed species index (**e**), Chao1 (**f**), PCoA (**g**), and NMDS (**h**) showed the α- and β-diversity of colonic microbiota. In Figure 4, GF0 (blue), basal diet group; GF4 (red), 0.4% GF group; GF6 (green), 0.6% GF group; GF8 (purple), 0.8% GF group.

**Figure 5 animals-15-00221-f005:**
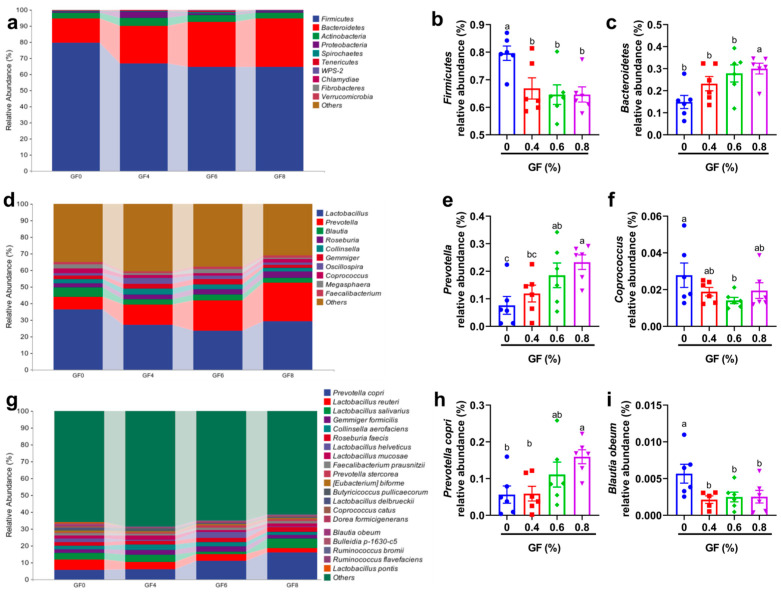
Effect of GF on relative abundance of colonic microbiota in piglets. ^a,b,c^ Means within a row with no common superscript differ significantly (*p* < 0.05) (n = 6). (**a**–**c**) the relative abundance of colonic microbiota at the phylum level; (**d**–**f**) the relative abundance of colonic microbiota at the genus level; (**g**–**i**) the relative abundance of colonic microbiota at the species level. In Figure 5a,d,g, GF0, basal diet group; GF4, 0.4% GF group; GF6, 0.6% GF group; GF8, 0.8% GF group. In Figure 5b,c,e,f,h,i, blue, basal diet group; red, 0.4% GF group; green, 0.6% GF group; purple, 0.8% GF group.

**Figure 6 animals-15-00221-f006:**
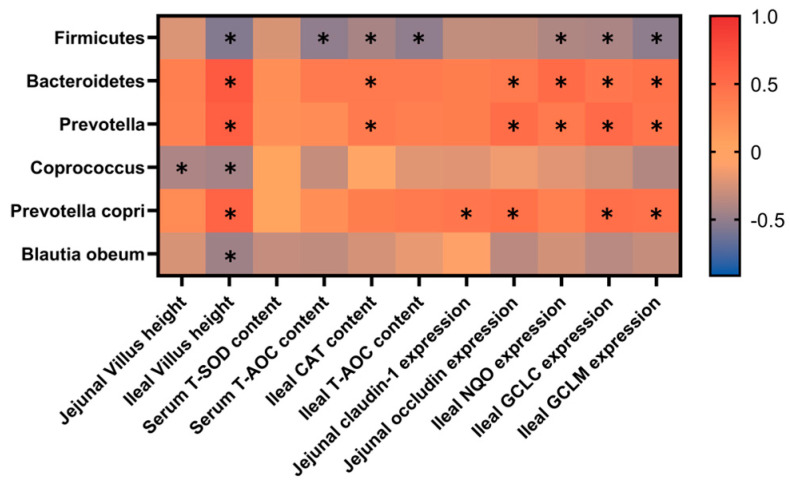
Correlation analysis based on significantly differential microbes and the performance. * indicates a significant difference at *p* < 0.05.

**Table 1 animals-15-00221-t001:** Formulation and chemical composition of the basal diet.

Ingredients	%	Nutrient Levels	
Corn	21.21	NE, kcal/kg	2600
Expanded corn	20.00	Crude protein, %	22.73
Soybean meal	19.46	SID ^2^ CP, %	18.43
Expanded soybean	11.00	SID Lys, %	1.43
Whey powder	10.00	SID Thr, %	0.84
Lactose	4.00	SID Trp, %	0.26
Fishmeal	3.00	SID met + cys, %	0.78
Soybean oil	2.82	Calcium, %	0.79
Sucrose	2.00	Total phosphorus, %	0.62
Soybean hull	2.00	STTD ^3^ P, %	0.40
Lysine HCl	0.33		
DL-Methionine	0.14		
L-Threonine	0.09		
Calcium hydrophosphate	0.80		
Limestone	1.00		
NaCl	0.35		
Monocalcium phosphate	0.18		
Titania	0.40		
Choline chloride	0.20		
Phytase	0.02		
Premix ^1^	1.00		
Total	100.00		

^1^ Provided vitamin and mineral premix per kg diet: vitamin A, 2400 IU; vitamin D_3_, 2800 IU; vitamin E, 200 IU; vitamin K_3_, 5 mg; vitamin B_12_, 40 μg; vitamin B_1_, 3 mg; vitamin B_2_, 10 mg; pantothenic acid, 15 mg; folic acid, 1 mg; vitamin B_6_, 8 mg; biotin, 0.08 mg; Fe (FeSO_4_·H_2_O), 120 mg; Cu (CuSO_4_·5H_2_O), 16 mg; Mn (MnSO_4_·H_2_O), 70 mg; Zn (ZnSO_4_·H_2_O), 120 mg; I (CaI_2_O_6_), 0.7 mg; and Se (Na_2_SeO_3_), 0.48 mg. ^2^ SID, Standardized ileal digestibility. ^3^ STTD, Standardized total tract digestible.

**Table 2 animals-15-00221-t002:** Primers used for quantitative real-time PCR ^1^.

Primer	Sequence (5′–3′)
*β-actin-F*	CATCGTCCACCGCAAAT
*β-actin-R*	TGTCACCTTCACCGTTCC
*SOD1-F*	GAGACCTGGGCAATGTGACT
*SOD1-R*	CTGCCCAAGTCATCTGGTTT
*GCLC-F*	CAAACCATCCTACCCTTTGG
*GCLC-R*	ATTGTGCAGAGAGCCTGGTT
*GCLM-F*	GATGCCGCCCGATTTAACTG
*GCLM-R*	ACAATGACCGAGTACCGCAG
*HO-1-F*	CGCTCCCGAATGAACACTCT
*HO-1-R*	GCGAGGGTCTCTGGTCCTTA
*NQO-F*	ATCACAGGTAAACTGAAGGA
*NQO-R*	TGGCAGCGTATGTGTAAGCA
*Nrf2-F*	CCCATTCACAAAAGACAAACA
*Nrf2-R*	GCTTTTGCCCTTAGCTCATCTC
*ZO-1-F*	AGCCCGAGGCGTGTTT
*ZO-1-R*	GGTGGGAGGATGCTGTTG
*Occludin-F*	GCACCCAGCAACGACAT
*Occludin-R*	CATAGACAGAATCCGAAT
*Claudin-1-F*	ACGGCCCAGGCCATCTAC
*Claudin-1-R*	TGCCGGGTCCGGTAGATG

^1^ SOD1, Superoxide Dismutase 1; GCLC, Glutamate-cysteine ligase catalytic subunit; GCLM, Glutamate-cysteine ligase catalytic; HO-1, Heme oxygenase-1; NQO, NAD(P)H: quinone oxidoreductase 1; Nrf2, Nuclear factor (erythroid-derived 2)-like 2; ZO-1, zonula occludens-1.

**Table 3 animals-15-00221-t003:** Effect of GF on growth performance of weaned piglets ^1^.

Items	GF (%)				SEM	*p*-Value			
	0	0.4	0.6	0.8		Anova	Linear	Quadratic	Cubic
Initial weight (kg)	8.28	8.29	8.28	8.32	0.12	0.907			
Final weight (kg)	19.99	20.29	19.70	20.63	0.23	0.219	0.530	0.667	0.560
ADG (g)	390.04	400.02	380.81	410.42	5.21	0.068	0.38	0.447	0.216
ADFI (g)	536.04	526.28	511.12	529.89	7.06	0.281	0.606	0.542	0.667
F/G	1.37 ^a^	1.32 ^ab^	1.35 ^ab^	1.29 ^b^	0.01	<0.05	0.056	0.168	0.111
Diarrhea rate (%)	13.99 ^a^	11.21 ^ab^	10.86 ^ab^	8.48 ^b^	0.69	<0.05	0.004	0.016	0.033

^1,a,b^ Means within a row with no common superscript differ significantly (*p* < 0.05) (n = 6). ADG, average daily gain; ADFI, average daily feed intake; F/G, feed conversion ratio.

**Table 4 animals-15-00221-t004:** Effect of GF on serum biochemical parameters of weaned piglets ^1^.

Items	GF (%)				SEM	*p*-Value			
	0	0.4	0.6	0.8		Anova	Linear	Quadratic	Cubic
CRE (μM/L)	101.45	106.77	103.61	99.21	2.95	0.667	0.702	0.655	0.824
UREA (mM/L)	3.34	4.43	3.55	4.17	0.18	0.305	0.355	0.646	0.433
GLU (mM/L)	4.40	4.44	4.55	4.28	0.15	0.986	0.867	0.874	0.950
TG (mM/L)	0.83	0.74	0.70	1.12	0.11	0.478	0.294	0.200	0.333
TP (g/L)	49.44	49.24	48.33	49.39	1.71	0.719	0.886	0.922	0.962
ALB (g/L)	25.07	26.58	26.26	27.30	0.23	0.627	0.209	0.452	0.585
HDL-C (mM/L)	0.74 ^b^	0.75 ^b^	0.88 ^a^	0.82 ^a^	0.02	0.014	0.067	0.146	0.111
LDL-C (mM/L)	0.83	1.06	1.12	1.05	0.04	0.137	0.097	0.070	0.157
DAO (ng/L)	258.18	207.75	161.11	191.36	18.88	0.106	0.147	0.199	0.341

^1,a,b^ Means within a row with no common superscript differ significantly (*p* < 0.05) (n = 6). CRE, creatinine; UREA, urea; GLU, glucose; TG, triglyceride; CHO, cholesterol; TP, total protein; ALB, albumin; HDL-C, high-density lipoprotein cholesterol; LDL-C, low-density lipoprotein cholesterol; DAO, diamine oxidase.

**Table 5 animals-15-00221-t005:** Effect of GF on serum immunological indicators of weaned piglets ^1^.

Items	GF (%)				SEM	*p*-Value			
	0	0.4	0.6	0.8		Anova	Linear	Quadratic	Cubic
IL-1β (ng/L)	18.55 ^ab^	15.93 ^b^	19.80 ^a^	18.03 ^ab^	0.35	<0.01	0.472	0.654	<0.01
IL-6 (ng/L)	302.65 ^b^	292.85 ^b^	441.03 ^ab^	485.33 ^a^	28.81	0.023	0.004	0.016	0.023
IL-10 (ng/L)	84.06 ^c^	105.41 ^b^	125.12 ^a^	137.28 ^a^	4.77	<0.01	< 0.01	< 0.01	<0.01
IL-22 (ng/L)	34.75	32.61	33.45	34.03	0.37	0.064	0.695	0.173	0.217
TNF-α (pg/mL)	288.92	286.43	281.80	268.61	5.38	0.245	0.179	0.366	0.576
TGF-β (pg/mL)	255.80	247.32	217.90	248.22	6.78	0.076	0.401	0.257	0.210
Ig A (μg/mL)	39.43 ^b^	44.06 ^ab^	41.87 ^ab^	47.96 ^a^	1.39	<0.05	0.059	0.168	0.165
Ig G (μg/mL)	203.03 ^b^	295.34 ^a^	240.74 ^ab^	228.36 ^ab^	12.59	0.039	0.854	0.110	0.054
Ig M (μg/mL)	22.02	21.98	22.05	23.09	0.29	0.504	0.22	0.318	0.504

^1,a,b,c^ Means within a row with no common superscript differ significantly (*p* < 0.05). IL-1β, Interleukin-1 beta; IL-6, Interleukin-6; IL-10, Interleukin-10; IL-22, Interleukin-22; TNF-α, Tumor necrosis factor-alpha; TGF-β, Transforming growth factor-beta; Ig A, Immunoglobulin A; Ig G, Immunoglobulin A; Ig M, Immunoglobulin A.

**Table 6 animals-15-00221-t006:** Effect of GF on antioxidant-related indicators in serum and intestine of weaned piglets ^1^.

Items	GF (%)				SEM	*p*-Value			
	0	0.4	0.6	0.8		Anova	Linear	Quadratic	Cubic
Serum									
MDA, nM/mL	239.74 ^a^	136.14 ^b^	228.83 ^a^	140.64 ^b^	15.30	0.008	0.114	0.270	0.008
T-AOC, mM/L	0.43 ^b^	0.51 ^a^	0.53 ^a^	0.49 ^a^	0.01	0.013	0.054	0.004	0.013
T-SOD, IU/mL	31.98 ^b^	37.02 ^a^	34.31 ^ab^	35.31 ^a^	0.58	0.006	0.147	0.049	0.006
CAT, IU/mL	39.06	30.32	48.71	46.99	2.98	0.103	0.117	0.234	0.103
GSH-Px, IU/L	1720.07	2137.05	1807.53	1993.92	74.76	0.186	0.446	0.518	0.186
Jejunum									
MDA, nM/mL	0.33 ^a^	0.19 ^b^	0.20 ^b^	0.17 ^b^	0.02	0.001	0.001	0.001	0.001
T-SOD, IU/mL	19.67	23.55	26.51	21.73	0.08	0.287	0.446	0.176	0.287
CAT, IU/mL	1.16	1.52	1.39	1.38	1.24	0.465	0.479	0.381	0.465
GSH-Px, IU/L	517.96 ^b^	584.16 ^b^	491.29 ^b^	740.23 ^a^	27.79	0.001	0.011	0.009	0.001
Ileum									
MDA, nM/mL	1.39 ^a^	0.89 ^b^	0.91 ^b^	0.91 ^b^	0.06	0.002	0.007	0.001	0.002
T-AOC, mM/L	0.61 ^c^	0.83 ^ab^	0.77 ^b^	0.96 ^a^	0.04	0.002	0.001	0.004	0.002
T-SOD, IU/mL	109.81 ^b^	123.49 ^a^	108.65 ^b^	99.38 ^b^	2.58	0.004	0.042	0.006	0.004
CAT, IU/mL	1.22 ^c^	1.57 ^ab^	1.39 ^b^	1.69 ^a^	0.06	0.007	0.010	0.037	0.007
GSH-Px, IU/L	35.24	43.43	41.05	41.34	1.80	0.156	0.333	0.350	0.436

^1,a,b,c^ Means within a row with no common superscript differ significantly (*p* < 0.05) (n = 6). MDA, Malondialdehyde; T-AOC, Total Antioxidant Capacity; T-SOD, Total Superoxide Dismutase; CAT, Catalase; GSH-Px, Glutathione Peroxidase.

**Table 7 animals-15-00221-t007:** Effect of GF on intestinal mucosal morphology of weaned piglets ^1^.

Items	GF (%)				SEM	*p*-Value			
	0	0.4	0.6	0.8		Anova	Linear	Quadratic	Cubic
Jejunum									
Villus height (μm)	407.99 ^b^	435.32 ^ab^	443.39 ^ab^	465.96 ^a^	34.72	0.049	0.029	0.097	0.192
Crypt depth (μm)	422.19	395.28	358.29	394.92	14.43	0.171	0.369	0.373	0.505
V/C	0.97	1.10	1.24	1.18	0.10	0.057	0.037	0.066	0.15
Ileum							0.001	0.004	0.007
Villus height (μm)	365.09 ^b^	387.45 ^b^	446.09 ^a^	440.92 ^a^	10.78	0.007	0.994	0.763	0.461
Crypt depth (μm)	327.83	317.39	364.72	314.25	11.64	0.179	0.023	0.068	0.137
V/C	1.11 ^b^	1.22 ^ab^	1.22 ^ab^	1.40 ^a^	0.05	<0.05	0.029	0.097	0.192

^1,a,b^ Means within a row with no common superscript differ significantly (*p* < 0.05) (n = 6). V/C, villus height/crypt depth ratio.

## Data Availability

The data presented in this study are available from the corresponding author upon reasonable request.

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
