# Peer review of "Effect of Gardeniae Fructus Powder on Growth Performance, Antioxidant Capacity, Intestinal Barrier Function, and Colonic Microbiota of Weaned Piglets"

_animals, 2025, doi:10.3390/ani15020221_

Round 1
Reviewer 1 Report (Previous Reviewer 3)
Comments and Suggestions for Authors
The concept of this submission is fine. However authors needs to improve the presentation of their data. Please find my below comments
In title, the word should be powder (not power), revise it in whole text.
Author did not follow the journal style in title setting, double check
L18-Powder
L19: compared with control or compared with basal diet.
Simple summary: authors should distinguish abstract and simple summary, revise simple summary focusing on only general importance of this study to readers.
L34-35: misleading statement, statistically it is not correct according to table. For example, IL-6 (table 5) in 0% group, 0.4% and 0.6% showed sub-letter b, so there were no different.
Similar misleading statement in whole text. HDLC in table 4.
I would like to suggest author to state specific difference in specific dosages to compare with control, Not a broad statement for all groups.
L36: Also misleading statement for MDA in serum; (check table 6) where o.6% showed higher and no difference between 05 and 0.6%.
My opinion to Revise abstract accordingly.
L38: For T-SOD and T-AOC > according to data table 6, T-AOC of serum were higher in all level of supplement, not only 0.8%?
I would like to suggest authors to represent their major findings accurately to avoid misleading according to analysis data.
The active component or the chemical composition/ nutritional analysis is missing for Gardeniae Fructus powder in this study.
Author Response
Response to Reviewer’s Comments
Dec 25, 2024,
Dear reviewer,
Thank you for comments concerning our manuscript entitled “Effect of Gardeniae Fructus Powder on Growth Performance, Antioxidant Capacity, Intestinal Barrier Function and Colonic Microbiota of Weaned Piglets” (Manuscript ID: animals- 3392100). These comments are valuable and helpful for revising the current manuscript and also for improving our further researches. We have studied comments carefully and have made the revisions in the manuscript, and we hope these revisions would meet with approval. Our responses to the comments are provided in the following text.
The concept of this submission is fine. However, authors needs to improve the presentation of their data. Please find my below comments
Comments 1. In title, the word should be powder (not power), revise it in whole text.
Response: We are sorry for our careless, “power” has been changed to “powder” in the manuscript.
Comments 2. Author did not follow the journal style in title setting, double check
Response: We are sorry for our careless, “Effect of Gardeniae Fructus powder on growth performance, antioxidant capacity, intestinal barrier function and colonic microbiota of weaned piglets” has been changed to “Effect of Gardeniae Fructus Powder on Growth Performance, Antioxidant Capacity, Intestinal Barrier Function and Colonic Microbiota of Weaned Piglets”.
”
Comments 3. L18-Powder
Response: We are sorry for our careless, “power” has been changed to “powder” in the manuscript.
Comments 4. L19: compared with control or compared with basal diet.
Response: Thanks for your suggestion, we have changed “compared with 0 %” to “compared with basal diet” in the manuscript.
Comments 5. Simple summary: authors should distinguish abstract and simple summary, revise simple summary focusing on only general importance of this study to readers.
Response: According to the reviewer’ s suggestion, we have revise simple summary to “Recent studies have demonstrated Gardeniae Fructus (GF), its extracts, and active components (Geniposide, genipin and chlorogenic acid et.al) alleviated colitis by improved intestinal antioxidant and immune capacity. However, little attention has been paid to the possible regulating effect and the underlying mechanism of GF powder on piglets suffering weaning stress. This study investigated the effects of GF powder on the growth performance, diarrhea rate, antioxidant and immune capacity, and intestinal health of weaned piglets. Results showed that GF supplementation at 0.8 % level significantly reduced feed / weight gain (F/G) and diarrhea rate compared with basal diet. GF supplementation also improved the abundance of Prevotella and Prevotella copri in the colon, increased antioxidant capacity and alleviates inflammatory response, ultimately maintaining gut barrier function. The work provides further insight into the beneficial effects of GF in swine production.” in the manuscript.
Comments 6. L34-35: misleading statement, statistically it is not correct according to table. For example, IL-6 (table 5) in 0% group, 0.4% and 0.6% showed sub-letter b, so there were no different.
Comments 7. Similar misleading statement in whole text. HDLC in table 4.
Response 6 and 7: Thanks for the reviewer’ s suggestion, in our results, in comparison with basal diet, 0.8 % GF significantly increased the contents of IL-6. Compared with basal diet, supplement with 0.6 and 0.8 % GF significantly increased HDL-C content in serum. Thus, we changed the description of these results to “Serum biochemical parameters showed that supplement with GF significantly increased the content of HDL-C (0.6 and 0.8 % level), IL-6(0.8 % level), IL-10 (0.4, 0.6 and 0.8 % level), Ig G (0.4 % level) and Ig A (0.8 % level) compared with basal diet (p<0.05)”
Comments 8. I would like to suggest author to state specific difference in specific dosages to compare with control, Not a broad statement for all groups.
Comments 9. L36: Also misleading statement for MDA in serum; (check table 6) where o.6% showed higher and no difference between 05 and 0.6%. My opinion to Revise abstract accordingly.
Comments 10. L38: For T-SOD and T-AOC > according to data table 6, T-AOC of serum were higher in all level of supplement, not only 0.8%? I would like to suggest authors to represent their major findings accurately to avoid misleading according to analysis data.
Response 8, 9 and 10: Thanks for the reviewer’ s suggestion, the description of index of antioxidant capacity has been changed in the abstract. “Index of antioxidant capacity showed that compared with basal diet, supplement with GF significantly decreased serum MDA content (0.4 % and 0.8 % level), jejunal and ileal MDA content (0.4 %, 0.6 % and 0.8 % level). Additionally, compared with basal diet, supplement with GF significantly increased serum and ileal T-AOC content (0.4 %, 0.6 % and 0.8 % level), serum T-SOD content (0.4 % and 0.8 % level), ileal T-SOD content (0.4 %, 0.6 % and 0.8 % level) and CAT content (0.4 %, 0.6 % and 0.8 % level), jejunal GSH-Px content (0.8 % level). The results of gene expression indicate that compared with basal diet, supplement with GF significantly increased Nrf 2 (0.4 % level), NQO (0.4 % level), SOD 1 (0.4 % and 0.8 % level), GCLC (0.4 % level) and GCLM (0.8 % level) abundance in jejunal mucosa, increased Nrf 2 (0.4 %, 0.6 % and 0.8 % level), HO-1 (0.4 % level), NQO (0.8 % level), SOD 1 (0.4 % and 0.8 % level), GCLC (0.4 % level) and GCLM (0.8 % level) abundance in ileal mucosa.”
Comments 11. The active component or the chemical composition/ nutritional analysis is missing for Gardeniae Fructus powder in this study.
Response: Thanks for the reviewer’ s suggestion. The dried mature fruit of Gardenia jasminoides Ellis. It is rarely elliptical, measuring 2.5-3.5 cm in length and 1-1.5cm in diameter, deep red or reddish yellow. The dried mature fruit was powdered using a pulverizer and filtered through a sieve (80 mesh), then stored at room temperature (25 °C) for use. The content of crude protein, crude fat, crude fiber, crude ash, calcium and total phosphorus were determined based on the conventional nutrient determination methods of feed ingredients adopt the national standard method of GB/T6432—2014, GB /T6433-2014, GB /T6434—2014, GB /T6438—2014, GB/T6436-2018 and GB/T6437-2018. The main active ingredients of the GF powder include total flavonoids, total phenol and soluble sugar were determined based on Plant flavonoids test kit (A142-1-1), Plant total phenol test kit (A143-1-1) and Plant soluble sugar content test kit (A145-1-1) (Nanjing Jiancheng Institute of Bioengineering, Nanjing, China). The nutritional levels and main bioactive compounds of GF powder were listed in Table S1.
Supplementary Table 1. The nutritional levels and main bioactive compounds of GF powder
|
Items |
Content |
|
Crude protein |
10.21 % |
|
Crude fat |
16.60 % |
|
Crude fiber |
23.20 % |
|
Crude ash |
3.70 % |
|
Calcium |
0.24 % |
|
Total phosphorus |
0.17 % |
|
Total flavonoids |
3.23 mg/g |
|
Total phenol |
1.2 μmol/g |
|
Soluble sugar |
6.50 mg/g |
Once again, we would like to express our great appreciation to you and reviewers for comments on our paper.
Looking forward to hearing from you soon.
Best wishes,
Shilong Liu
Institute of Animal Science, Guangdong Academy of Agricultural Science, 510640 Guangzhou, Guangdong, China. E-mail: liushilong94@126.com.
Phone/Fax: +86-20-85161287
Reviewer 2 Report (Previous Reviewer 2)
Comments and Suggestions for Authors
This manuscript investigated the Effects of Gardeniae Fructus power on growth performance, antioxidant capacity, Intestinal barrier function and colonic microbiota of weaned piglets. There are no problems with the experimental design. The author has detected many indicators and the experiment was done very carefully. However, there is not much introduction to the research object, Gardeniae Fructus powder. There is no data on its preparation method and main active ingredients, which affects the reproducibility of the research. In addition, the description of relevant materials and methods is not sufficient, and details need to be added. There is also room for improvement in the analysis of experimental data. My suggestions are as follows:
Line 125-126: How was this Gardeniae Fructus powder added to the feed? Was the feed pellet feed or powder feed?
Line 146-148: Did the author analyze the main active ingredients of the Gardeniae Fructus powder? What was their content? Or how was this Gardeniae Fructus powder prepared? What were its product standards or requirements? Which part of Gardeniae fructus was selected and crushed for the test materials? Only by clarifying these can the research method in this study be replicated.
Line 156: “one piglet chosen from each pen” a gilt or barrow?
Line 256-257: The statistical method used was one-way ANOVA. Were factors such as gender, pig pen, and initial body weight considered? It is recommended to use a mixed model to analyze the growth data, with treatment and gender as fixed factors and pig pen as a random factor.
Table 3, 4, 5, 6, 7: Since there are four concentration gradients in this study, it is recommended to add the p-values of linear, quadratic, and cubic to evaluate whether there is a regression trend in these indicators with the increase of GF concentration.
Figure 6: Here, it is actually the correlation analysis of two groups of variables, that is, the correlation between the indicators of intestinal morphology, antioxidant and gene expression and the abundance of multiple intestinal microorganisms. It is recommended to use canonical correlation, which is used to analyze the correlation of two groups of variables. Because there are correlations among these variables, analyzing the correlation of two variables separately without considering the influence of other variables on these two variables will especially bring bias in small sample analysis.
Author Response
Response to Reviewer’s Comments
Dec 25, 2024,
Dear reviewer,
Thank you for comments concerning our manuscript entitled “Effect of Gardeniae Fructus Powder on Growth Performance, Antioxidant Capacity, Intestinal Barrier Function and Colonic Microbiota of Weaned Piglets” (Manuscript ID: animals- 3392100). These comments are valuable and helpful for revising the current manuscript and also for improving our further researches. We have studied comments carefully and have made the revisions in the manuscript, and we hope these revisions would meet with approval. Our responses to the comments are provided in the following text.
This manuscript investigated the Effects of Gardeniae Fructus power on growth performance, antioxidant capacity, Intestinal barrier function and colonic microbiota of weaned piglets. There are no problems with the experimental design. The author has detected many indicators and the experiment was done very carefully. However, there is not much introduction to the research object, Gardeniae Fructus powder. There is no data on its preparation method and main active ingredients, which affects the reproducibility of the research. In addition, the description of relevant materials and methods is not sufficient, and details need to be added. There is also room for improvement in the analysis of experimental data. My suggestions are as follows:
Comments 1. Line 125-126: How was this Gardeniae Fructus powder added to the feed? Was the feed pellet feed or powder feed?
Response: Thanks for the reviewer’ s suggestion. Firstly, when preparing the premix, mix the GF powder and premix evenly, and then, mixture of premix and GF powder is pelletized together with raw materials such as corn and soybean meal.
Comments 2. Line 146-148: Did the author analyze the main active ingredients of the Gardeniae Fructus powder? What was their content? Or how was this Gardeniae Fructus powder prepared? What were its product standards or requirements? Which part of Gardeniae fructus was selected and crushed for the test materials? Only by clarifying these can the research method in this study be replicated.
Response: Thanks for the reviewer’ s suggestion. The dried mature fruit of Gardenia jasminoides Ellis. It is rarely elliptical, measuring 2.5-3.5 cm in length and 1-1.5cm in diameter, deep red or reddish yellow. The dried mature fruit was powdered using a pulverizer and filtered through a sieve (80 mesh), then stored at room temperature (25 °C) for use. The content of crude protein, crude fat, crude fiber, crude ash, calcium and total phosphorus were determined based on the conventional nutrient determination methods of feed ingredients adopt the national standard method of GB/T6432—2014, GB /T6433-2014, GB /T6434—2014, GB /T6438—2014, GB/T6436-2018 and GB/T6437-2018. The main active ingredients of the GF powder include total flavonoids, total phenol and soluble sugar were determined based on Plant flavonoids test kit (A142-1-1), Plant total phenol test kit (A143-1-1) and Plant soluble sugar content test kit (A145-1-1) (Nanjing Jiancheng Institute of Bioengineering, Nanjing, China). The nutritional levels and main bioactive compounds were listed in Table S1
Supplementary Table 1. The nutritional levels and main bioactive compounds of GF powder
|
Items |
Content |
|
Crude protein |
10.21 % |
|
Crude fat |
16.60 % |
|
Crude fiber |
23.20 % |
|
Crude ash |
3.70 % |
|
Calcium |
0.24 % |
|
Total phosphorus |
0.17 % |
|
Total flavonoids |
3.23 mg/g |
|
Total phenol |
1.2 μmol/g |
|
Soluble sugar |
6.50 mg/g |
Comments 3 Line 156: “one piglet chosen from each pen” a gilt or barrow?
Response: Thanks for the reviewer’ s suggestion, gilt or barrow is random. one piglet chosen from each pen based on average body weight. The selected pig is just close to the average weight in this pen.
Comments 4. Line 256-257: The statistical method used was one-way ANOVA. Were factors such as gender, pig pen, and initial body weight considered? It is recommended to use a mixed model to analyze the growth data, with treatment and gender as fixed factors and pig pen as a random factor.
Comments 5. Table 3, 4, 5, 6, 7: Since there are four concentration gradients in this study, it is recommended to add the p-values of linear, quadratic, and cubic to evaluate whether there is a regression trend in these indicators with the increase of GF concentration.
Response 4 and 5: Thanks for the reviewer’ s suggestion. As suggested by the reviewer, gender, pig pen, and initial body weight were need be considered. But, firstly, our team's research on 7-25 kg piglets shows that gender had no significant effect on growth performance (unpublished data), this is the foundation that helps us design experiments. And then, 3 barrows and 3 gilts were raised in the same pen, feed intake is measured in units of pens, so the feed intake and F/G of barrows and gilts cannot be accurately calculated. Thus, Not considering gender factors maybe better reflect data on growth performance. And in future studies, we will pay more attention to these factors when designing experiments, thanks again. According to the reviewer’ s suggestion. Growth and other data shown in Table 3-7, we have added the linear, quadratic, and cubic analysis to theses tables in the manuscript.
Table 3. Effect of GF on growth performance of weaned piglets 1.
|
Items |
GF (%) |
|
|
|
SEM |
p-value |
|
|
|
|
|
0 |
0.4 |
0.6 |
0.8 |
|
Anova |
Linear |
Quadratic |
Cubic |
|
Initial weight (kg) |
8.28 |
8.29 |
8.28 |
8.32 |
0.12 |
0.907 |
|
|
|
|
Final weight (kg) |
19.99 |
20.29 |
19.70 |
20.63 |
0.23 |
0.219 |
0.530 |
0.667 |
0.560 |
|
ADG (g) |
390.04 |
400.02 |
380.81 |
410.42 |
5.21 |
0.068 |
0.38 |
0.447 |
0.216 |
|
ADFI (g) |
536.04 |
526.28 |
511.12 |
529.89 |
7.06 |
0.281 |
0.606 |
0.542 |
0.667 |
|
F/G |
1.37a |
1.32ab |
1.35ab |
1.29b |
0.01 |
<0.05 |
0.056 |
0.168 |
0.111 |
|
Diarrhea rate (%) |
13.99a |
11.21ab |
10.86ab |
8.48b |
0.69 |
<0.05 |
0.004 |
0.016 |
0.033 |
1 a, b Means within a row with no common superscript differ significantly (p < 0.05) (n=6). ADG, average daily gain; ADFI, average daily feed intake; F/G, feed conversion ratio.
Table 4. Effect of GF on serum biochemical parameters of weaned piglets 1.
|
Items |
GF (%) |
|
|
|
SEM |
p-value |
|
|
|
|
|
0 |
0.4 |
0.6 |
0.8 |
|
Anova |
Linear |
Quadratic |
Cubic |
|
CRE |
101.45 |
106.77 |
103.61 |
99.21 |
2.95 |
0.667 |
0.702 |
0.655 |
0.824 |
|
UREA |
3.34 |
4.43 |
3.55 |
4.17 |
0.18 |
0.305 |
0.355 |
0.646 |
0.433 |
|
GLU |
4.40 |
4.44 |
4.55 |
4.28 |
0.15 |
0.986 |
0.867 |
0.874 |
0.950 |
|
TG |
0.83 |
0.74 |
0.70 |
1.12 |
0.11 |
0.478 |
0.294 |
0.200 |
0.333 |
|
TP |
49.44 |
49.24 |
48.33 |
49.39 |
1.71 |
0.719 |
0.886 |
0.922 |
0.962 |
|
ALB |
25.07 |
26.58 |
26.26 |
27.30 |
0.23 |
0.627 |
0.209 |
0.452 |
0.585 |
|
HDL-C |
0.74b |
0.75b |
0.88a |
0.82a |
0.02 |
0.014 |
0.067 |
0.146 |
0.111 |
|
LDL-C |
0.83 |
1.06 |
1.12 |
1.05 |
0.04 |
0.137 |
0.097 |
0.070 |
0.157 |
|
DAO |
258.18 |
207.75 |
161.11 |
191.36 |
18.88 |
0.106 |
0.147 |
0.199 |
0.341 |
1 a, b Means within a row with no common superscript differ significantly (p < 0.05) (n=6). CRE, creatinine; UREA, urea; GLU, glucose; TG, triglyceride; CHO, cholesterol; TP, total protein; ALB, albumin; HDL-C, high-density lipoprotein cholesterol; LDL-C, low-density lipoprotein cholesterol; DAO, diamine oxidase.
Table 5. Effect of GF on serum immunological indicators of weaned piglets 1.
|
Items |
GF (%) |
|
|
|
SEM |
p-value |
|
|
|
|
|
0 |
0.4 |
0.6 |
0.8 |
|
Anova |
Linear |
Quadratic |
Cubic |
|
IL-1β (ng/L) |
18.55ab |
15.93b |
19.80a |
18.03ab |
0.35 |
< 0.01 |
0.472 |
0.654 |
< 0.01 |
|
IL-6 (ng/L) |
302.65b |
292.85b |
441.03ab |
485.33a |
28.81 |
0.023 |
0.004 |
0.016 |
0.023 |
|
IL-10 (ng/L) |
84.06c |
105.41b |
125.12a |
137.28a |
4.77 |
< 0.01 |
< 0.01 |
< 0.01 |
< 0.01 |
|
IL-22 (ng/L) |
34.75 |
32.61 |
33.45 |
34.03 |
0.37 |
0.064 |
0.695 |
0.173 |
0.217 |
|
TNF-α (pg/mL) |
288.92 |
286.43 |
281.80 |
268.61 |
5.38 |
0.245 |
0.179 |
0.366 |
0.576 |
|
TGF-β (pg/mL) |
255.80 |
247.32 |
217.90 |
248.22 |
6.78 |
0.076 |
0.401 |
0.257 |
0.210 |
|
Ig A (μg/mL) |
39.43b |
44.06ab |
41.87ab |
47.96a |
1.39 |
< 0.05 |
0.059 |
0.168 |
0.165 |
|
Ig G (μg/mL) |
203.03b |
295.34a |
240.74ab |
228.36ab |
12.59 |
0.039 |
0.854 |
0.110 |
0.054 |
|
Ig M (μg/mL) |
22.02 |
21.98 |
22.05 |
23.09 |
0.29 |
0.504 |
0.22 |
0.318 |
0.504 |
|
IL-1β (ng/L) |
18.55ab |
15.93b |
19.80a |
18.03ab |
0.35 |
< 0.01 |
0.472 |
0.654 |
< 0.01 |
1 a, b Means within a row with no common superscript differ significantly (p < 0.05). IL-1β, Interleukin-1 beta; IL-6, Interleukin-6; IL-10, Interleukin-10; IL-22, Interleukin-22; TNF-α, Tumor necrosis factor-alpha; TGF-β, Transforming growth factor-beta; Ig A, Immunoglobulin A; Ig G, Immunoglobulin A; Ig M, Immunoglobulin A.
Table 6. Effect of GF on antioxidant-related indicators in serum and intestine of weaned piglets1.
|
Items |
GF (%) |
|
|
|
SEM |
p-value |
|
|
|
|
|
0 |
0.4 |
0.6 |
0.8 |
|
Anova |
Linear |
Quadratic |
Cubic |
|
Serum |
|
|
|
|
|
|
|
|
|
|
MDA, nM/mL |
239.74a |
136.14b |
228.83a |
140.64b |
15.30 |
0.008 |
0.114 |
0.270 |
0.008 |
|
T-AOC, mM/L |
0.43b |
0.51a |
0.53a |
0.49a |
0.01 |
0.013 |
0.054 |
0.004 |
0.013 |
|
T-SOD, IU/mL |
31.98b |
37.02a |
34.31ab |
35.31a |
0.58 |
0.006 |
0.147 |
0.049 |
0.006 |
|
CAT, IU/mL |
39.06 |
30.32 |
48.71 |
46.99 |
2.98 |
0.103 |
0.117 |
0.234 |
0.103 |
|
GSH-Px, IU/L |
1720.07 |
2137.05 |
1807.53 |
1993.92 |
74.76 |
0.186 |
0.446 |
0.518 |
0.186 |
|
Jejunum |
|
|
|
|
|
|
|
|
|
|
MDA, nM/mL |
0.33a |
0.19b |
0.20b |
0.17b |
0.02 |
0.001 |
0.001 |
0.001 |
0.001 |
|
T-SOD, IU/mL |
19.67 |
23.55 |
26.51 |
21.73 |
0.08 |
0.287 |
0.446 |
0.176 |
0.287 |
|
CAT, IU/mL |
1.16 |
1.52 |
1.39 |
1.38 |
1.24 |
0.465 |
0.479 |
0.381 |
0.465 |
|
GSH-Px, IU/L |
517.96b |
584.16b |
491.29b |
740.23a |
27.79 |
0.001 |
0.011 |
0.009 |
0.001 |
|
Ileum |
|
|
|
|
|
|
|
|
|
|
MDA, nM/mL |
1.39a |
0.89b |
0.91b |
0.91b |
0.06 |
0.002 |
0.007 |
0.001 |
0.002 |
|
T-AOC, mM/L |
0.61c |
0.83ab |
0.77b |
0.96a |
0.04 |
0.002 |
0.001 |
0.004 |
0.002 |
|
T-SOD, IU/mL |
109.81b |
123.49a |
108.65b |
99.38b |
2.58 |
0.004 |
0.042 |
0.006 |
0.004 |
|
CAT, IU/mL |
1.22c |
1.57ab |
1.39b |
1.69a |
0.06 |
0.007 |
0.010 |
0.037 |
0.007 |
|
GSH-Px, IU/L |
35.24 |
43.43 |
41.05 |
41.34 |
1.80 |
0.156 |
0.333 |
0.350 |
0.436 |
1 a, b Means within a row with no common superscript differ significantly (p < 0.05) (n=6). MDA, Malondialdehyde; T-AOC, Total Antioxidant Capacity; T-SOD, Total Superoxide Dismutase; CAT, Catalase; GSH-Px, Glutathione Peroxidase.
Table 7. Effect of GF on intestinal mucosal morphology of weaned piglets 1.
|
Items |
GF (%) |
|
|
|
SEM |
p-value |
|
|
|
|
|
0 |
0.4 |
0.6 |
0.8 |
|
Anova |
Linear |
Quadratic |
Cubic |
|
Jejunum |
|
|
|
|
|
|
|
|
|
|
Villus height (μm) |
407.99b |
435.32ab |
443.39ab |
465.96a |
34.72 |
0.049 |
0.029 |
0.097 |
0.192 |
|
Crypt depth (μm) |
422.19 |
395.28 |
358.29 |
394.92 |
14.43 |
0.171 |
0.369 |
0.373 |
0.505 |
|
V/C |
0.97 |
1.10 |
1.24 |
1.18 |
0.10 |
0.057 |
0.037 |
0.066 |
0.15 |
|
Ileum |
|
|
|
|
|
|
0.001 |
0.004 |
0.007 |
|
Villus height (μm) |
365.09b |
387.45b |
446.09a |
440.92a |
10.78 |
0.007 |
0.994 |
0.763 |
0.461 |
|
Crypt depth (μm) |
327.83 |
317.39 |
364.72 |
314.25 |
11.64 |
0.179 |
0.023 |
0.068 |
0.137 |
|
V/C |
1.11b |
1.22ab |
1.22ab |
1.40a |
0.05 |
<0.05 |
0.029 |
0.097 |
0.192 |
1 a, b Means within a row with no common superscript differ significantly (p < 0.05) (n=6). V/C, villus height /crypt depth ratio.
Comments 6. Figure 6: Here, it is actually the correlation analysis of two groups of variables, that is, the correlation between the indicators of intestinal morphology, antioxidant and gene expression and the abundance of multiple intestinal microorganisms. It is recommended to use canonical correlation, which is used to analyze the correlation of two groups of variables. Because there are correlations among these variables, analyzing the correlation of two variables separately without considering the influence of other variables on these two variables will especially bring bias in small sample analysis.
Response: Thanks for the reviewer’ s suggestion. As shown in Figure 6, the canonical correlation analysis between significantly differential colonic microbiota and antioxidant related indicators, intestinal epithelium functions. Our results showed that ileal villus height, serum T-AOC content, ileal CAT and T-AOC content, expressions of NQO, GCLC and GCLM were negatively associated with the abundance of Firmicutes (phylum) (p < 0.05). Jejunal and ileal villus height were negatively associated with the abundance of Coprococcus (genus) (p < 0.05). Ileal villus height was negatively associated with the abundance of Blautia obeum (species) (p < 0.05). Ileal villus height, ileal CAT content, expressions of jejunal occluding, ileal NQO, GCLC and GCLM were positively associated with the abundance of Bacteroidetes (phylum) and Prevotella (genus) (p < 0.05). Ileal villus height, expressions of jejunal claudin-1 and occludin, ileal GCLC and GCLM were positively associated with the abundance of Prevotella copri (species) (p < 0.05).
Figure 6. The correlation analysis based on between significantly differential microbes and the performance. * indicate a significant difference at p < 0.05.
Once again, we would like to express our great appreciation to you and reviewers for comments on our paper.
Looking forward to hearing from you soon.
Best wishes,
Shilong Liu
Institute of Animal Science, Guangdong Academy of Agricultural Science, 510640 Guangzhou, Guangdong, China. E-mail: liushilong94@126.com.
Phone/Fax: +86-20-85161287

Reviewer 3 Report (New Reviewer)
Comments and Suggestions for Authors
Animals - MDPI
Manuscript Number: 3392100
Overview: Authors propose utilization of an GF extract powder on growth performance, antioxidant capacity, intestinal barrier function, and colonic microbiota of weaned piglets.
Overall Comments: Overall, I believe that this manuscript presents a convincing argument as to the positive effects of including GF in the diets of weaned piglets on a variety of parameters. Upon these minor revisions and some English revisions, I recommend that this manuscript be published.
At first I thought the Gardeniae Fructus power was just a one-time mistype, but it is throughout the whole manuscript, including the title. Please make sure that this is corrected throughout. I couldn’t make this make sense in my head, because I assume you mean powder (how the plant material was supplemented to the basal diets.
Section and Line-by-Line Comments:
Abstract:
18: powder? Make sure to check this throughout manuscript, EVEN THE TITLE
Introduction/Background:
77: why is Ellis not italicized, it seems out of place.
Materials and Methods:
123-124: A total of 144 (genetic line) weaned piglets (8.29+/- 0.11 kg; 21 d) were randomly assigned to 4 treatments.
124: You should introduce the diets/treatments here and explain their relative differences.
126: supplemented
Results:
296: make sure to put a space; “in the four groups”; check this throughout manuscript
Discussion/Conclusion:
In regards to cost-benefit, is GF a reasonable additive to propose to producers?
Tables and Figures:
No major comments
References: No major comments here.
Comments on the Quality of English Language
There are just some inconsistencies with sentence structure and verbiage that should addressed throughout.
Author Response
Response to Reviewer’s Comments
Dec 25, 2024,
Dear reviewer,
Thank you for comments concerning our manuscript entitled “Effect of Gardeniae Fructus Powder on Growth Performance, Antioxidant Capacity, Intestinal Barrier Function and Colonic Microbiota of Weaned Piglets” (Manuscript ID: animals- 3392100). These comments are valuable and helpful for revising the current manuscript and also for improving our further researches. We have studied comments carefully and have made the revisions in the manuscript, and we hope these revisions would meet with approval. Our responses to the comments are provided in the following text.
Overview: Authors propose utilization of an GF extract powder on growth performance, antioxidant capacity, intestinal barrier function, and colonic microbiota of weaned piglets.
Overall Comments: Overall, I believe that this manuscript presents a convincing argument as to the positive effects of including GF in the diets of weaned piglets on a variety of parameters. Upon these minor revisions and some English revisions, I recommend that this manuscript be published.
Comments 1. At first I thought the Gardeniae Fructus power was just a one-time mistype, but it is throughout the whole manuscript, including the title. Please make sure that this is corrected throughout. I couldn’t make this make sense in my head, because I assume you mean powder (how the plant material was supplemented to the basal diets.
Response: We are sorry for our careless, “power” has been changed to “powder” in the manuscript.
Section and Line-by-Line Comments:
Abstract:
Comments 2. 18: powder? Make sure to check this throughout manuscript, EVEN THE TITLE
Response: We are sorry for our careless, “power” has been changed to “powder” in the manuscript.
Introduction/Background:
Comments 3. 77: why is Ellis not italicized, it seems out of place.
Response: Thanks for the reviewer’ s suggestion, according to the International Code of Nomenclature for Algae, Fungi, and Plants (ICN), the scientific names of plants need to be written in italics, while the names of namers (such as J. Ellis) do not need to be italicized.
Materials and Methods:
Comments 4. 123-124: A total of 144 (genetic line) weaned piglets (8.29+/- 0.11 kg; 21 d) were randomly assigned to 4 treatments. You should introduce the diets/treatments here and explain their relative differences.
Response: Thanks for the reviewer’ s suggestion, we have changed this sentence to “A total 144 [Duroc × (Landrace × Yorkshire)] weaned piglets (8.29±0.11 kg) with 21 d old randomly assigned to 4 groups, include a basal diet supplemented with 0%, 0.4%, 0.6% and 0.8% Gardeniae Fructus powder, respectively (n=36). Each treatment consisted of 6 replicate pens, with 6 piglets per pen and each pen containing 3 barrows and 3 gilts.” in the manuscript.
Comments 5. 126: supplemented
Response: We are sorry for our careless, “supplement” has been changed to “supplemented” in our manuscript.
Comments 6. 296: make sure to put a space; “in the four groups”; check this throughout manuscript
Response: We are sorry for our careless, “in fourgroups” has been changed to “in four groups” in the manuscript.
Discussion/Conclusion:
Comments 7. In regards to cost-benefit, is GF a reasonable additive to propose to producers?
Response: Thanks for the reviewer’ s suggestion, the price of GF is similar to that of soybean hull. Thus, it is suitable for adding to feed.
Tables and Figures:
No major comments
References: No major comments here.
Once again, we would like to express our great appreciation to you and reviewers for comments on our paper.
Looking forward to hearing from you soon.
Best wishes,
Shilong Liu
Institute of Animal Science, Guangdong Academy of Agricultural Science, 510640 Guangzhou, Guangdong, China. E-mail: liushilong94@126.com.
Phone/Fax: +86-20-85161287

Round 2
Reviewer 1 Report (Previous Reviewer 3)
Comments and Suggestions for Authors
Dear Author
Thanks for revising ur submission. i would like to suggest authors to further improvement of this submission, specially authors needs to provide scientific justification against their major findings, Please check the attached file as example.
In addition, language should be improve in whole text.

needs to improve,
Author Response
Response to Reviewer’s Comments
Jan. 05, 2025,
Dear reviewer,
Thank you for comments concerning our manuscript entitled “Effect of Gardeniae Fructus Powder on Growth Performance, Antioxidant Capacity, Intestinal Barrier Function and Colonic Microbiota of Weaned Piglets” (Manuscript ID: animals- 3392100). These comments are valuable and helpful for revising the current manuscript and also for improving our further researches. We have studied comments carefully and have made the revisions in the manuscript, and we hope these revisions would meet with approval. Our responses to the comments are provided in the following text.
Comments 1. Recent studies have demonstrated Gardeniae Fructus (GF), its extracts, and ac- 15
tive components (geniposide, genipin and chlorogenic acid et.al) alleviated colitis by improved in-
testinal antioxidant and immune capacity.
Response: According to the reviewer’ s suggestion, we have changed “improved” to “improving” in our manuscript.
Comments 2. Table 4. mention the unit for each test parameters
Response: We are sorry for our careless, we have added unit for each test parameters to Table 4 in our manuscript.
Table 4. Effect of GF on serum biochemical parameters of weaned piglets 1.
|
Items |
GF (%) |
|
|
|
SEM |
p-value |
|
|
|
|
|
0 |
0.4 |
0.6 |
0.8 |
|
Anova |
Linear |
Quadratic |
Cubic |
|
CRE (μM/L) |
101.45 |
106.77 |
103.61 |
99.21 |
2.95 |
0.667 |
0.702 |
0.655 |
0.824 |
|
UREA (mM/L) |
3.34 |
4.43 |
3.55 |
4.17 |
0.18 |
0.305 |
0.355 |
0.646 |
0.433 |
|
GLU (mM/L) |
4.40 |
4.44 |
4.55 |
4.28 |
0.15 |
0.986 |
0.867 |
0.874 |
0.950 |
|
TG (mM/L) |
0.83 |
0.74 |
0.70 |
1.12 |
0.11 |
0.478 |
0.294 |
0.200 |
0.333 |
|
TP (g/L) |
49.44 |
49.24 |
48.33 |
49.39 |
1.71 |
0.719 |
0.886 |
0.922 |
0.962 |
|
ALB (g/L) |
25.07 |
26.58 |
26.26 |
27.30 |
0.23 |
0.627 |
0.209 |
0.452 |
0.585 |
|
HDL-C (mM/L) |
0.74b |
0.75b |
0.88a |
0.82a |
0.02 |
0.014 |
0.067 |
0.146 |
0.111 |
|
LDL-C (mM/L) |
0.83 |
1.06 |
1.12 |
1.05 |
0.04 |
0.137 |
0.097 |
0.070 |
0.157 |
|
DAO (ng/L) |
258.18 |
207.75 |
161.11 |
191.36 |
18.88 |
0.106 |
0.147 |
0.199 |
0.341 |
Comments 3. There is not justification on how the treatment is effective to reduce diarrhea and growth? Authors need to explain their findings (Line 419-420, 422-429).
Response: Thanks for your suggestion, we have rewrote the discussion line (419-429). “Weaning stress in piglets typically manifests itself as decreased growth performance and an increased incidence of diarrhea immediately after abrupt weaning[23,24]. The gene and protein expression of tight junction proteins (Claudin-1, Occludin, and ZO-1) are significantly reduced after weaning of piglets[25]. Zhu et al. found that weaning stress led to an increase in serum levels of nitric oxide and hydrogen peroxide, a decrease in expression of antioxidant enzymes in the jejunum, and damage to intestinal villi in piglets[26]. Previous studies have shown that plant extracts (such as Guizhi Li-Zhong extract, Huangqin decoction, Codonopsis Pilosulae and Astragalus Membranaceus extract) can alleviate oxidative stress, improve immunity capacity, increase beneficial bacteria abundance, and improve intestinal barrier function, thereby alleviating weaning stress in piglets[27-29]. A study by Liu et al. indicated that Gardenia jasminoides Ellis fruit extracts relieved oxidative stress and inflammatory reaction thereby alleviated TNBS-induced colitis in rats[18]. In the present study, our results indicated that dietary supplement with Gardeniae Fructus powder at a level of 0.8 % significantly decreased piglets’ diarrhea rate and F/G after a 30-day feeding trial. At the same time, there is an increasing trend for ADG in the 0.8 % GF group. And supplement with Gardeniae Fructus powder (0.8 %) significantly improved the levels of T-AOC, T-SOD, CAT, IL-10 and decreased MDA in serum and small intestine compared with control group, thereby increased villus height in ileum. And the indices related to intestinal barrier function showed GF significantly increased expression of ZO-1, Occludin and Claudin-1, which may be associate with improvement of Prevotella and Prevotella copri relative abundance in piglets. Thus, in the present study, GF reduce diarrhea rate and F/G maybe by increasing intestinal antioxidant levels, villus height, and intestinal barrier function in piglets.”
Comments 4. Not meaningful sentence/not complete “Key contributing factors include the transitioning from milk to solid feed, altered nutrient composition, an immature digestive system, and increased bad gut bacteria[25].” (Line 420-422)
Response: Thanks for your suggestion, this sentence has been deleted.
Comments 5. Supplement with GF had no significant effect on ADFI compared to piglets fed a 0.basal diet diet. (Line 429)
Response: We are sorry for our careless, this sentence has been changed to “Supplement with GF had no significant effect on ADFI compared to piglets fed a basal diet.”
Comments 6. Rewrite “Thus, the effective effect of GF on growth performance suggests an improvement in intestinal health in weaned piglets.” (Line 429-430)
Response: Thanks for your suggestion, this sentence has been changed to “Thus, in the present study, our results indicated that GF reduce diarrhea rate and F/G maybe by increasing intestinal antioxidant levels, villus height, and intestinal barrier function in piglets.”
Comments 7. Serum biochemical indicators serve as crucial tools for assessing an organism's health status, providing valuable insights into the functioning of vital organs and metabolic processes. These indicators can reflect the function and metabolic state of multiple organ systems.
Response: Thanks for your suggestion, these two sentences have repeated meanings and we have deleted “These indicators can reflect the function and metabolic state of multiple organ systems.” in the manuscript.
Comments 5. The present results showed supplement with GF at 0.8 % level significantly increased villus height in jejunum, villus height and villus/crypt ratio in ileum compared with basal diet (p=0.057). In ileum, compared with basal diet. This suggests that GF improved the digestive ability of the small intestine.
Response: We are sorry for our careless, “In ileum, compared with basal diet” has been deleted in the manuscript.
Comments 6. occludin and Claudin-1 in the GF groups.
Response: We are sorry for our careless, “occludin” has been changed to “Occludin” in the manuscript.
Once again, we would like to express our great appreciation to you and reviewers for comments on our paper.
Looking forward to hearing from you soon.
Best wishes,
Shilong Liu
Institute of Animal Science, Guangdong Academy of Agricultural Science, 510640 Guangzhou, Guangdong, China. E-mail: liushilong94@126.com.
Phone/Fax: +86-20-85161287

Round 3
Reviewer 1 Report (Previous Reviewer 3)
Comments and Suggestions for Authors
Thanks for necessary modification
Comments on the Quality of English LanguagePlease revise the language by related expert one
This manuscript is a resubmission of an earlier submission. The following is a list of the peer review reports and author responses from that submission.
Round 1
Reviewer 1 Report
Comments and Suggestions for Authors
In this study, the authors investigated the effect of Gardeniae Fructus on growth performance and intestinal health of weaned piglets, and demonstrated that Gardeniae Fructus effectively decreased diarrhea rate, improved antioxidant capacity and intestinal barrier function. This study is of interest and will give us new information, especially in the field of feed additive.
Minor points for revisions are as follows:
Abstract
1. Please provide gender details for these piglets.
2. Please check and address the grammar and sentence structure throughout the manuscript.
3. Bacterial names should be in italics.
M&M
4. Line 122, add the replicates for each group.
5. Authors need to provide essential information regarding the rearing conditions of piglets. Such as where were allotted piglet? Cages or floors?
6. Similarly, how did the authors feed the piglets daily? How was the water provided? Ad libitum? Please add it.
7. The authors need to provide source information for the Gardeniae Fructus used in the experiment.
8. How did the authors sacrifice animals? This is a very important, please add it.
9. “The basal diet (control) was formulated to meet the nutrient recommendations of the National Research Council 2012 (NRC, 2012)” Lack of references. Please add it.
Results
10. Line 285-287. Please provide these genes description.
11. Line 299-302. In addition to statistical values, H&E staining of the intestine also important to reflect intestinal development. It is suggested that the authors supplement the results with H&E staining.
12. The authors have presented the results of oxidative / antioxidant enzyme activities in serum, liver tissue, and jejunal tissue in Tables 5, 6, and 8 respectively. It is suggested that the authors combine these parts of the results.
Discussion
13. Line 459-498. Bacterial names should be in italics. Please check it.
References
14. Some references in the article are incomplete (such as references 1, 2 and 11, the page numbers are incomplete).
Author Response
Reply is attached.

Reviewer 2 Report
Comments and Suggestions for Authors
Line 115: Please indicate the breed, genetic background and sex of these piglets. Weaning on the 21st day? How to wean? Did these piglets enter the experiment immediately after weaning or was there a pre-trial period? There were only 20 pigs in each group, and the sample size was small.
Line 119-120:Please explain: Where were these pigs raised? How was the pen divided? How many pigs were there in a pen? What was the area of each pen? What facilities were there? What were the feeding conditions? What were the environmental control conditions of the pig barn? What equipment was installed? Because all these feeding and management conditions are related to these performance indicators of pigs.
Line 123: Please explain in detail how the diarrhea rate of piglets was calculated.
Line 172: How many pigs were slaughtered? How many pig samples were collected in each group?
Line 204-220: How many pigs were tested here? Only 5? What platform was used for 16S RNA sequencing analysis? In which biotechnology company? Please explain.
Line 223: It is too simple to use only one-way ANOVA. You didn't consider other factors?
Table 2: The statistical power depends on several factors such as the effect size, sample size, significance level (p-value), and variability of the data. With only 60 observations and considering the mean and standard deviation for these production performance indicators, it is unlikely that the statistical power would reach 80% or above for all the parameters at a p-value of 0.05. For example, if the power for Final weight is calculated to be only 0.36, it indicates that there is a relatively low probability of detecting a true effect if it exists. A power of less than 80% suggests that the study may not be adequately powered to detect significant differences for some of the parameters, and the results should be interpreted with caution. Increasing the sample size or reducing the variability of the data could potentially improve the power of the statistical tests.
Table 7: What’s means these “FCR”? If this refers to the VCR, then this data is problematic. Please check again.
Figure 4: for figure 4g and 4h,
Figure 5: There are only 15 samples, right? Please show the results of each pig in each group for Figure 5a, 5c, and 5e. The reviewers want to see the differences within the group and the differences within the group.
Figure 6: In Figure 6, in the pairwise simple correlation analysis between the abundance of each microorganism and each measured index, how to eliminate the interference of a third factor in this small sample data? Because when you analyze the correlation between the abundance of a certain microorganism and one of the indexes, the abundances of other microorganisms are not consistent, and there will also be interference among other indexes. Therefore, using canonical correlation or partial correlation analysis would be better.
Author Response
Reply is attached.

Reviewer 3 Report
Comments and Suggestions for Authors
Dear author thanks for your good submission in Animals J. The concept of this study is fine. The methodology is correct. Data was analyzed correctly.
Results was organized and well presented BUT some findings are contradictory (L322-25).
Discussion part needs to improve more. I have below comments:
L50-delete >which may be associated from .
Gardeniae Fructus > fructus or Fructus?
L67> delete healthy
L79: >Chlorogenic 79 acid is present in high content in organic acids[16], what is the relationship?
L114-123: Please provide information regarding Gardeniae Fructus, source, form, mixing procedure in diets.
Result table 3: LDLC was higher in treatment group that not good for health. Anyway authors needs to explains in text properly.
L267> delete .
How many sample was taken (n=?) need to mention in each table footnote
L322-325: Fig. 3. b; How those findings good for health?, usually higher of those parameters (IL-1β, TGF-β) are negative for good health., authors needs to explain in text.
L386-392: those are repeated statement, u can write in shortly but main things needs to justify ur own findings. why and how the treatment group affects on major findings. Revise accordingly in every significant results.
L400-408: similarly u need to explains the logical reasons of changes the findings.
L422-423: According to Rf no.35, your findings and statement are contradictory, Here some justification is needed.
L483: As ref. 62 , your findings L322-24 (TGF-B) are contradictory,
Author Response
Reply is attached.
